# Lung Cancer Inequalities in Stage of Diagnosis in Ontario, Canada

Aisha K. Lofters [1,2,3,4,5,*] , Evgenia Gatov [4] , Hong Lu [4], Nancy N. Baxter [4,6,7,8], Sara J. T. Guilcher [4,9] , Alexander Kopp [4], Mandana Vahabi [4,10] and Geetanjali D. Datta [11,12,13]

1 Department of Family & Community Medicine, University of Toronto, Toronto, ON M5G 1V7, Canada
2 Women's College Hospital Research Institute, Toronto, ON M5S 1B2, Canada
3 Peter Gilgan Centre for Women's Cancers, Women's College Hospital, Toronto, ON M5S 1B2, Canada
4 ICES, 2075 Bayview Ave, Toronto, ON M4N 3M5, Canada; jenny.gatov@mail.utoronto.ca (E.G.); hong.lu@ices.on.ca (H.L.); nancy.baxter@unimelb.edu.au (N.N.B.); Sara.guilcher@utoronto.ca (S.J.T.G.); Alexander.kopp@ices.on.ca (A.K.); mvahabi@ryerson.ca (M.V.)
5 MAP Centre for Urban Health Solutions, St. Michael's Hospital, Toronto, ON M5B 1W8, Canada
6 Melbourne School of Population and Global Health, University of Melbourne, Victoria 3053, Australia
7 Department of Surgery, University of Toronto, Toronto, ON M5T 1P5, Canada
8 Li Ka Shing Knowledge Institute, St. Michael's Hospital, Toronto, ON M5B 1W8, Canada
9 Leslie Dan Faculty of Pharmacy, University of Toronto, Toronto, ON M5S 3M2, Canada
10 Daphne Cockwell School of Nursing, Ryerson University, Toronto, ON M5B 1Z5, Canada
11 Department of Social and Preventive Medicine, Université de Montréal, Montréal, QC H3N 1X9, Canada; Geetanjali.Datta@cshs.org
12 Research Center of the University of Montreal Hospital Center (CR-CHUM), Montréal, QC H2X 0A9, Canada
13 Cancer Research Center for Health Equity, Cedars-Sinai Medical Center, Los Angeles, CA 90069, USA
* Correspondence: aisha.lofters@wchospital.ca

**Abstract:** Lung cancer is the most common cancer and cause of cancer death in Canada, with approximately 50% of cases diagnosed at stage IV. Sociodemographic inequalities in lung cancer diagnosis have been documented, but it is not known if inequalities exist with respect to immigration status. We used multiple linked health-administrative databases to create a cohort of Ontarians 40–105 years of age who were diagnosed with an incident lung cancer between 1 April 2012 and 31 March 2017. We used modified Poisson regression with robust standard errors to examine the risk of diagnosis at late vs. early stage among immigrants compared to long-term residents. The fully adjusted model included age, sex, neighborhood-area income quintile, number of Aggregated Diagnosis Group (ADG) comorbidities, cancer type, number of prior primary care visits, and continuity of care. Approximately 62% of 38,788 people with an incident lung cancer from 2012 to 2017 were diagnosed at a late stage. Immigrants to the province were no more likely to have a late-stage diagnosis than long-term residents (63.5% vs. 62.0%, relative risk (RR): 1.01 (95% confidence interval (CI): 0.99–1.04), adjusted relative risk (ARR): 1.02 (95% CI: 0.99–1.05)). However, in fully adjusted models, people with more comorbidities were less likely to have a late-stage diagnosis (adjusted relative risk (ARR): 0.82 (95% CI: 0.80–0.84) for those with 10+ vs. 0–5 ADGs). Compared to adenocarcinoma, small cell carcinoma was more likely to be diagnosed at a late stage (ARR: 1.29; 95% CI: 1.27–1.31), and squamous cell (ARR: 0.89; 95% CI: 0.87–0.91) and other lung cancers (ARR: 0.93; 95% CI: 0.91–0.94) were more likely to be diagnosed at an early stage. Men were also slightly more likely to have late-stage diagnosis in the fully adjusted model (ARR: 1.08; 95% CI: 1.05–1.08). Lung cancer in Ontario is a high-fatality cancer that is frequently diagnosed at a late stage. Having fewer comorbidities and being diagnosed with small cell carcinoma was associated with a late-stage diagnosis. The former group may have less health system contact, and the latter group has the lung cancer type most closely associated with smoking. As lung cancer screening programs start to be implemented across Canada, targeted outreach to men and to smokers, increasing awareness about screening, and connecting every Canadian with primary care should be system priorities.

**Keywords:** lung cancer; screening; immigrant health

## 1. Introduction

Lung cancer is currently the most common cancer and the most common cause of cancer death in Canada, with approximately 30,000 new diagnoses and 20,000 deaths in 2020 [1]. Unfortunately, 50% of lung cancer cases in Canada currently are diagnosed at stage IV [1]. This late-stage diagnosis is particularly concerning as stage I survival is 71% versus 5% for those diagnosed at stage IV [1]. While prevention through reducing tobacco use is still the key to reducing lung cancer incidence and mortality, a larger knowledge base about where, how, and with whom to intervene in the healthcare system is needed for earlier diagnosis of lung cancer at the population level.

Current literature on social inequalities in lung cancer stage of diagnosis suggests that certain demographic groups may be appropriate target populations for interventions. Canadians with low incomes have been found to be more likely to be diagnosed with lung cancer at a late stage than those with high income; the same holds true for Canadian men versus women [1]. In the US, Nadpara et al. found longer times from symptoms to lung cancer diagnosis for non-White patients [2]. However, despite the fact that 22% of Canadians are foreign-born, according to the 2016 Canadian Census [3], little is known about if inequalities exist in lung cancer diagnosis for Canadian immigrants. This is a potentially important gap in the literature, particularly in provinces like Ontario, Canada's most populous province, where nearly 30% of the population are immigrants [3].

In this population-level retrospective cohort study, we used Ontario health and administrative databases to compare the stage of diagnosis (dichotomized as early vs. late) of lung cancer for immigrants to Ontario versus long-term residents of the province and to examine the association of stage of diagnosis with potentially actionable sociodemographic and health-related factors.

## 2. Materials and Methods

### 2.1. Data Sources

We used several databases available at ICES. ICES is an independent, non-profit research institute funded by an annual grant from the Ontario Ministry of Health and the Ministry of Long-Term Care. As a prescribed entity under Ontario's privacy legislation, ICES is authorized to collect and use health care data for the purposes of health system analysis, evaluation, and decision support. Secure access to these data is governed by policies and procedures that are approved by the Information and Privacy Commissioner of Ontario. ICES houses a secure array of Ontario's health-related administrative data. Data include population demographics and health service use information on all Ontario residents who are eligible for the province's universal Ontario Health Insurance Plan (OHIP). All datasets are linked using unique encoded identifiers and analyzed at ICES.

The Immigration, Refugee and Citizenship Canada Permanent Resident (IRCC) database consists of detailed demographic information on Ontario's immigrants and refugees who landed from 1985 onward, including country of birth, date of landing, and immigrant class. The Ontario Cancer Registry (OCR) documents data on all Ontario residents who have been newly diagnosed with or died of cancer (except non-melanoma skin cancers), including the primary cancer site, diagnosis date, age at diagnosis, stage at diagnosis, histology, and cause of death if applicable. To determine the stage at diagnosis, Collaborative Staging is used where available [4]. The Collaborative Staging System is a unified data collection system used in Canada and the United States and is based on a set of pathological and clinical data items, including tumor size, extension, lymph node status, and metastasis status [4]. If Collaborative Stage is not available, the stage is based on physician staging from the regional cancer center. If neither is captured (due to limited stage work-up and/or limited documentation in the patient's medical record), the stage is coded as missing [5]. The capture of Collaborative Stage for lung cancer became available at ICES in 2010, and as of 2013, 90% of lung cancers in the OCR have complete staging information [5].

Other databases that we used included the Registered Persons Database (RPDB), which includes date of birth, sex, postal code, dates of contact with the healthcare system

and date of death if applicable; the OHIP Database, which contains procedural and diagnostic codes claimed by physicians in the province; the Client Agency Program Enrolment (CAPE) database which is updated bi-monthly and identifies all Ontarians who are enrolled with a family physician in Ontario's various primary care models [6]; the Corporate Provider Database (CDPB), which records which family physicians participate in these models; the ICES Physicians' Database (IPDB), which records demographic information about Ontario's physicians who are in active practice.

### 2.2. Study Cohort

We included all people 18–105 years of age residing in Ontario who were diagnosed with an incident lung cancer as per the OCR at any time between 1 April 2012 and 31 March 2017 and who had a unique ICES identifier. We excluded those whose cancer was stage 0 or in situ at the time of diagnosis, who had no Ontario healthcare system contact for the three years prior to diagnosis, and whose eligibility for provincial health insurance was after their diagnosis date. As more than 90% of Canada's immigrants live in urban settings [7], we excluded people who lived in rural areas, and finally, excluded those who were diagnosed with lung cancer at less than 40 years of age as this group may have a distinct disease process. We categorized the cohort into "immigrants" and "long-term residents" based on their inclusion in the IRCC database. The language of "long-term residents" was used as immigrants who arrived before 1985 would be included in this group.

### 2.3. Variables

For each patient, we documented sex, age, and postal code at the time of diagnosis from the RPDB. We used the postal code to identify the income quintile for their neighborhood based on the 2016 Canadian Census mean household income data. We used Aggregated Diagnosis Groups (ADGs) from the Johns Hopkins ACG® case-mix system (V10.0) to categorize comorbidity [8]. This case-mix system identifies comorbidities from diagnosis codes in outpatient billing and inpatient hospital records. The ADGs represent groups of conditions with similar healthcare experiences based on factors such as disease severity and duration. We categorized cancer types as small cell, squamous cell, adenocarcinoma, and other. To describe primary care contact, we determined the number of visits to a primary care physician for each patient in the 6 to 30 months before diagnosis (a two-year period) using OHIP data. We excluded primary care visits during the 0 to 6 months before diagnosis because these visits may reflect a peri-diagnostic interval as opposed to usual care [9]. We also used these visits to describe patients' continuity of care with a primary care physician based on the Usual Provider of Care (UPC) index, which measures the proportion of all primary care visits that were made to the provider most frequently visited, among those with at least three primary care visits in a two-year period [10]. High continuity of care is defined as a UPC index of at least 75%. We subcategorized immigrants by regions of origin, immigrant class, and by length of time in Canada at the date of diagnosis. To determine the region of origin, we used country of birth and categorized each country into a world region based on a modification of the World Bank classification previously published by AL [11].

### 2.4. Outcome

Outcomes were dichotomous: early (I–II) vs. late (III–IV) stage diagnosis based on the OCR.

### 2.5. Analysis

We described the cohort, overall and by immigrant status, on the basis of the variables noted above. We further stratified each of these groups by early vs. late stage of diagnosis and again described them based on the variables noted above. We used chi-square tests and standardized differences (preferable for large sample sizes) for group comparisons and considered a standardized difference of > 0.10 to be clinically important [12].

We used modified Poisson regression with robust standard errors to examine the risk of late vs. early stage of diagnosis among immigrants compared to long-term residents, first unadjusted, then age and sex-adjusted (age as a continuous variable), and then fully adjusted for age, sex, neighborhood-area income quintile, number of ADG comorbidities, cancer type, number of prior primary care visits, and UPC index. We removed cohort members with missing outcome data from the analysis. We also stratified each model by age category, by sex, and by age × sex category. We conducted statistical analyses using SAS software (version 9.4, Cary, NC, USA). The study was approved by the Research Ethics Board at Unity Health Toronto.

## 3. Results

Figure 1 describes the study cohort creation, and Table 1 describes the 38,788 individuals in the cohort, 2696 (7.0%) of whom were immigrants. The immigrant group was more likely to be male than long-term residents (58.7% vs. 50.1%) and had a younger median age at diagnosis (68 vs. 72 years). They were also more likely to be in the lowest neighborhood income quintile. Long-term residents had slightly more comorbidities (median of 8 vs. 7), but no notable differences were seen in primary care contact or continuity of care. Immigrants were more likely to be diagnosed with adenocarcinoma (49.6% vs. 37.3%) and were less likely to have small cell or squamous cell carcinoma. More than half of the immigrant group arrived in Ontario as sponsored families, 42.2% were from East Asia and the Pacific, and immigrants had been in Canada for a median of 19 years.

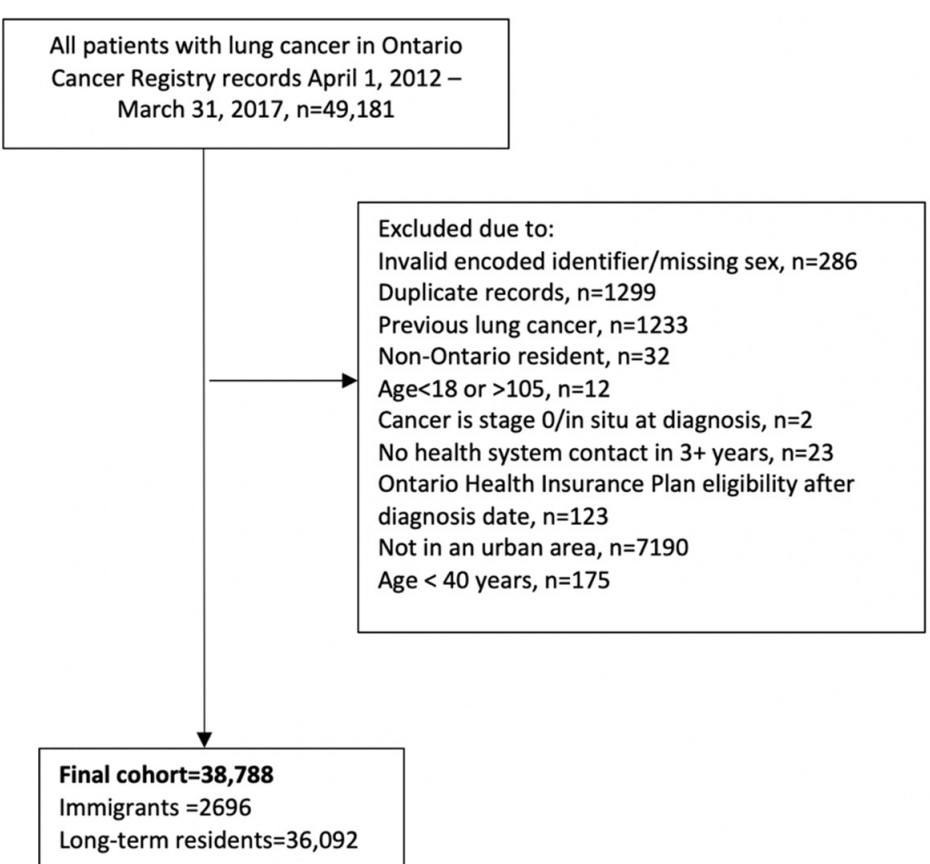

**Figure 1.** Creation of study cohort of 38,788 patients diagnosed with lung cancer in Ontario, 1 April 2012–31 March 2017.

**Table 1.** Descriptive characteristics by immigrant status of 38,788 people in Ontario diagnosed with lung cancer, 1 April 2012–31 March 2017.

| Characteristics | Immigrants (*n* = 2696) | Long-Term Residents (*n* = 36,092) | Standardized Difference | Total (*n* = 38,788) |
|---|---|---|---|---|
| **Sex** | | | | |
| Female | 1114 (41.3%) | 18,020 (49.9%) | 0.17 | 19,134 (49.3%) |
| Male | 1582 (58.7%) | 18,072 (50.1%) | 0.17 | 19,654 (50.7%) |
| **Age at diagnosis** | | | | |
| Mean ± SD | 67.73 ± 12.19 | 71.46 ± 10.53 | 0.33 | 71.20 ± 10.69 |
| Median (IQR) | 68 (59–77) | 72 (64–79) | 0.3 | 72 (64–79) |
| **Age group (years)** | | | | |
| 40–64 | 1105 (41.0%) | 9479 (26.3%) | 0.32 | 10,584 (27.3%) |
| 65–74 | 718 (26.6%) | 12,056 (33.4%) | 0.15 | 12,774 (32.9%) |
| 75+ | 873 (32.4%) | 14,557 (40.3%) | 0.17 | 15,430 (39.8%) |
| **Neighborhood income quintile** | | | | |
| Quintile 1 (lowest) | 824 (30.6%) | 8840 (24.5%) | 0.14 | 9664 (24.9%) |
| Q2 | 584 (21.7%) | 8178 (22.7%) | 0.02 | 8762 (22.6%) |
| Q3 | 488 (18.1%) | 6927 (19.2%) | 0.03 | 7415 (19.1%) |
| Q4 | 464 (17.2%) | 6261 (17.3%) | 0 | 6725 (17.3%) |
| Q5 (highest) | 333 (12.4%) | 5852 (16.2%) | 0.11 | 6185 (15.9%) |
| **Number of John's Hopkins ADG comorbidities** | | | | |
| Mean ± SD | 7.64 ± 3.55 | 8.16 ± 3.77 | 0.14 | 8.12 ± 3.76 |
| Median (IQR) | 7 (5–10) | 8 (5–11) | 0.14 | 8 (5–11) |
| 0–5 | 779 (28.9%) | 9179 (25.4%) | 0.08 | 9958 (25.7%) |
| 6–9 | 1156 (42.9%) | 14,246 (39.5%) | 0.07 | 15,402 (39.7%) |
| 10+ | 761 (28.2%) | 12,667 (35.1%) | 0.15 | 13,428 (34.6%) |
| **Lung cancer type** | | | | |
| Adenocarcinoma | 1336 (49.6%) | 13,478 (37.3%) | 0.25 | 14,814 (38.2%) |
| Small cell | 340 (12.6%) | 6158 (17.1%) | 0.13 | 6498 (16.8%) |
| Squamous cell | 363 (13.5%) | 6169 (17.1%) | 0.1 | 6532 (16.8%) |
| Other | 657 (24.4%) | 10,287 (28.5%) | 0.09 | 10,944 (28.2%) |
| **No. PCP visits 6–30 months < index—all primary care providers** | | | | |
| Mean ± SD | 9.41 ± 8.50 | 8.95 ± 8.67 | 0.05 | 8.98 ± 8.66 |
| Median (IQR) | 8 (4–13) | 7 (3–12) | 0.08 | 7 (3–12) |
| **No. PCP visits 6–30 months < index—patient's usual provider of care** | | | | |
| Mean ± SD | 6.91 ± 7.07 | 6.89 ± 7.38 | 0 | 6.89 ± 7.36 |
| Median (IQR) | 5 (1–10) | 5 (2–10) | 0.01 | 5 (2–10) |
| **Usual Provider of Care (UPC) index** | | | | |
| 0 visits | 240 (8.9%) | 3438 (9.5%) | 0.02 | 3678 (9.5%) |
| 1–2 visits | 307 (11.4%) | 4898 (13.6%) | 0.07 | 5205 (13.4%) |
| UPC <= 75% | 593 (22.0%) | 6522 (18.1%) | 0.1 | 7115 (18.3%) |
| UPC > 75% | 1556 (57.7%) | 21,234 (58.8%) | 0.02 | 22,790 (58.8%) |

ADG: Adjusted Diagnosis Groups; PCP: primary care provider.

For the overall cohort, 26.5% were diagnosed at an early stage, 62.2% were diagnosed at a late stage, and 11.3% were missing stage. Immigrants and long-term residents did not significantly differ by stage of diagnosis: 25.7%, 63.5%, and 10.8% of immigrants were diagnosed early, late, or had missing stage data, respectively, versus 26.6%, 62.0%, and

11.4% of long-term residents. In bivariate analyses (Table 2), for both groups, people diagnosed at an early stage were more likely to be women, with only 36.8% of immigrants diagnosed at a late stage being women. Both immigrants and long-term residents diagnosed at a late stage had fewer comorbidities and were more likely to have small cell carcinoma. For both groups, being diagnosed at an early stage was associated with slightly more contact with primary care prior to diagnosis versus their peers diagnosed at a late stage (immigrants: median of 7 vs. 5 visits to usual provider in the two years before diagnosis; long-term residents: median of 6 vs. 5 visits). Among immigrants, 21.6% of those diagnosed at a late stage had less than three visits to primary care in the two years prior to diagnosis vs. 15.2% of those diagnosed at an early stage. The stage of diagnosis was not associated with immigrant class, region of origin, or years in Canada (data not shown).

**Table 2.** Descriptive characteristics of study cohort stratified by stage of diagnosis (early vs. late) and immigrant status.

| Characteristics | Immigrants | | | Long-Term Residents | | |
|---|---|---|---|---|---|---|
| | Early Stage (*n* = 692) | Late Stage (*n* = 1713) | Standardized Difference | Early Stage (*n* = 9595) | Late Stage (*n* = 22,394) | Standardized Difference |
| Sex | | | | | | |
| Female | 341 (49.3%) | 630 (36.8%) | 0.25 | 5141 (53.6%) | 10,759 (48.0%) | 0.11 |
| Male | 351 (50.7%) | 1083 (63.2%) | 0.25 | 4454 (46.4%) | 11,635 (52.0%) | 0.11 |
| Age at diagnosis | | | | | | |
| Mean ± SD | 67.77 ± 11.40 | 66.84 ± 12.05 | 0.08 | 71.25 ± 9.75 | 70.70 ± 10.43 | 0.05 |
| Median (IQR) | 68 (59–77) | 67 (58–76) | 0.07 | 72 (65–78) | 71 (63–78) | 0.06 |
| Age group (years) | | | | | | |
| 40–64 | 274 (39.6%) | 746 (43.5%) | 0.08 | 2359 (24.6%) | 6424 (28.7%) | 0.09 |
| 65–74 | 195 (28.2%) | 460 (26.9%) | 0.03 | 3478 (36.2%) | 7573 (33.8%) | 0.05 |
| 75+ | 223 (32.2%) | 507 (29.6%) | 0.06 | 3758 (39.2%) | 8397 (37.5%) | 0.03 |
| Neighborhood income quintile | | | | | | |
| Quintile 1 (lowest) | 216 (31.2%) | 527 (30.8%) | 0.01 | 2280 (23.8%) | 5511 (24.6%) | 0.02 |
| Q2 | 152 (22.0%) | 378 (22.1%) | 0 | 2117 (22.1%) | 5115 (22.8%) | 0.02 |
| Q3 | 124 (17.9%) | 289 (16.9%) | 0.03 | 1839 (19.2%) | 4301 (19.2%) | 0 |
| Q4 | 98 (14.2%) | 321 (18.7%) | 0.12 | 1735 (18.1%) | 3861 (17.2%) | 0.02 |
| Q5 (highest) | 100 (14.5%) | 197 (11.5%) | 0.09 | 1619 (16.9%) | 3583 (16.0%) | 0.02 |
| No. John's Hopkins ADG comorbidities | | | | | | |
| Mean ± SD | 8.39 ± 3.43 | 7.25 ± 3.43 | 0.33 | 8.88 ± 3.60 | 7.77 ± 3.74 | 0.3 |
| Median (IQR) | 8 (6–11) | 7 (5–9) | 0.32 | 9 (6–11) | 8 (5–10) | 0.31 |
| 0–5 | 138 (19.9%) | 559 (32.6%) | 0.29 | 1706 (17.8%) | 6503 (29.0%) | 0.27 |
| 6–9 | 308 (44.5%) | 743 (43.4%) | 0.02 | 3878 (40.4%) | 8899 (39.7%) | 0.01 |
| 10+ | 246 (35.5%) | 411 (24.0%) | 0.25 | 4011 (41.8%) | 6992 (31.2%) | 0.22 |
| Lung cancer type | | | | | | |
| Adenocarcinoma | 380 (54.9%) | 887 (51.8%) | 0.06 | 4036 (42.1%) | 8788 (39.2%) | 0.06 |
| Small cell | 24 (3.5%) | 300 (17.5%) | 0.47 | 639 (6.7%) | 5319 (23.8%) | 0.49 |
| Squamous cell | 100 (14.5%) | 241 (14.1%) | 0.01 | 2290 (23.9%) | 3628 (16.2%) | 0.19 |
| Other | 188 (27.2%) | 285 (16.6%) | 0.26 | 2630 (27.4%) | 4659 (20.8%) | 0.15 |
| No. PCP visits 6–30 months < index—all primary care providers | | | | | | |
| Mean ± SD | 10.47 ± 7.85 | 8.92 ± 8.54 | 0.19 | 9.89 ± 8.32 | 8.59 ± 8.74 | 0.15 |
| Median (IQR) | 9 (5–14) | 7 (3–12) | 0.27 | 8 (4–13) | 6 (3–12) | 0.22 |

**Table 2.** *Cont.*

| Characteristics | Immigrants | | | Long-Term Residents | | |
|---|---|---|---|---|---|---|
| | Early Stage (*n* = 692) | Late Stage (*n* = 1713) | Standardized Difference | Early Stage (*n* = 9595) | Late Stage (*n* = 22,394) | Standardized Difference |
| No. PCP visits 6–30 months < index—patient's usual provider of care | | | | | | |
| Mean ± SD | 7.65 ± 6.57 | 6.52 ± 7.03 | 0.17 | 7.67 ± 7.34 | 6.63 ± 7.32 | 0.14 |
| Median (IQR) | 7 (3–11) | 5 (1–9) | 0.24 | 6 (2–11) | 5 (1–9) | 0.19 |
| UPC index | | | | | | |
| 0 visits | 45 (6.5%) | 159 (9.3%) | 0.1 | 687 (7.2%) | 2123 (9.5%) | 0.08 |
| 1–2 visits | 60 (8.7%) | 211 (12.3%) | 0.12 | 1103 (11.5%) | 3195 (14.3%) | 0.08 |
| UPC <= 75% | 167 (24.1%) | 380 (22.2%) | 0.05 | 1788 (18.6%) | 4010 (17.9%) | 0.02 |
| UPC > 75% | 420 (60.7%) | 963 (56.2%) | 0.09 | 6017 (62.7%) | 13,066 (58.3%) | 0.09 |

ADG: Adjusted Diagnosis Groups; PCP: primary care provider.

In regression analyses, immigrants had no differences in stage of diagnosis than long-term residents (Table 3). People with more comorbidities were less likely to have a late-stage diagnosis (adjusted relative risk (ARR): 0.82 (95% confidence interval (CI) 0.80–0.84) for those with 10+ vs. 0–5 ADGs). Cancer type was significantly associated with diagnostic stage in the fully adjusted model: compared to adenocarcinoma, small cell carcinoma was more likely to be diagnosed late-stage (ARR 1.29; 95% CI 1.27–1.31) and squamous cell (ARR 0.89; 95% CI 0.87–0.91) and other cancers (ARR 0.93; 95% CI 0.91–0.94) were more likely to be diagnosed at an early stage. Men were also slightly more likely to have a late-stage diagnosis in the fully adjusted model (ARR 1.08; 95% CI 1.05–1.08), but income quintile and continuity of care were not associated with diagnostic stage. No notable differences were seen for models stratified by age category, by sex, or by age category × sex. Post hoc, we repeated the models excluding people diagnosed with small cell carcinoma, and no notable differences were seen.

**Table 3.** Results from the multivariable model using Poisson regression. Adjusted relative risks represent late vs. early stage of diagnosis.

| Variables | Relative Risk (95% Confidence Interval) |
|---|---|
| Unadjusted | |
| Immigrant (vs. long term resident) | 1.02 (0.99–1.05) |
| Age and sex-adjusted | |
| Immigrant (vs. long term resident) | 1.00 (0.98–1.05) |
| Male (vs. female) | 1.08 (1.06–1.09) |
| Age (as a continuous variable) | 1.00 (1.00–1.00) |
| Full model | |
| Immigrant (vs. long term resident) | 1.01 (0.99–1.04) |
| Male (vs. female) | 1.07 (1.05–1.08) |
| Age (as a continuous variable) | 1.00 (1.00–1.00) |
| Neighborhood income quintile (quintile 5 as the reference group) | |
| Income quintile 1 (lowest) | 1.02 (1.00–1.04) |
| Income quintile 2 | 1.02 (0.99–1.04) |
| Income quintile 3 | 1.01 (0.99–1.03) |
| Income quintile 4 | 1.01 (0.98–1.03) |

**Table 3.** *Cont.*

| Variables | Relative Risk (95% Confidence Interval) |
|---|---|
| Comorbidities (0–5 ADGs as the reference group) | |
| 6–9 ADGs | 0.89 (0.88–0.91) |
| 10 + ADGs | 0.82 (0.80–0.84) |
| Primary care visits in the 6–30 months prior to diagnosis (as a continuous variable) | 1.00 (1.00–1.00) |
| Continuity of care (Usual Provider of Care Index of 75% or greater as the reference group) | |
| 0 visits to primary care | 1.02 (1.00–1.05) |
| 1–2 visits to primary care | 1.02 (1.00–1.04) |
| Usual Provider of Care Index less than 75% | 1.02 (1.01–1.04) |
| Lung cancer type (adenocarcinoma as the reference group) | |
| Small cell | 1.29 (1.27–1.31) |
| Squamous cell | 0.89 (0.87–0.91) |
| Other | 0.93 (0.91–0.94) |

## 4. Discussion

In this population-based retrospective cohort study in urban areas of Ontario, Canada, we found that more than 62% of people with an incident lung cancer from 2012 to 2017 were diagnosed at a late stage. In the context of a universal healthcare system, immigrants to the province were no more likely to have a late-stage diagnosis than long-term residents, and the diagnostic stage was not associated with immigrant class, region of origin, or years in Canada. However, in fully adjusted models, late-stage diagnosis was associated with male sex, fewer comorbidities, and being diagnosed with small cell carcinoma.

Our finding that the majority of lung cancers in Ontario are diagnosed late highlights the urgent need for a diagnostic stage shift. Lung cancer is typically diagnosed symptomatically in the Canadian context, and both the lack of specific symptoms and lack of awareness of symptoms can contribute to a delay in diagnosis [13,14]. Organized lung cancer screening programs do not currently exist across Canada, although one is currently being piloted across several regions in Ontario with plans for provincial expansion, and one is set to begin in British Columbia in 2022 [15,16]. Organized screening programs using low-dose CT scans and focussing on those at highest risk due to age and smoking history may lead to substantial improvements in lung cancer morbidity and mortality [17]. The Ontario lung screening pilot eligibility criteria are being ages 55 to 74 years, being a current or former daily smoker for at least 20 years in total, and PLCm201 risk of 2% or greater in the next six years [15]. Of note, data published from the Ontario lung screening pilot show that, to date, women have been more likely to participate than men [15]. Combined with our findings and national data [1], this gender difference suggests that screening programs may benefit from targeted recruitment and marketing efforts towards screen-eligible men in addition to broader awareness campaigns.

In fully adjusted models, we found that people with more comorbid conditions had a diagnostic stage advantage. In bivariate analyses, having more primary care contact in the two years prior to diagnosis was also associated with a stage advantage. We have previously found similar findings in Ontario for the stage of diagnosis for colorectal cancer, another prevalent cancer with high mortality [18], and other studies have shown that more comorbidities are associated with a higher likelihood of cancer screening and of dyslipidemia testing [19–21]. An increased number of comorbidities may lead to increased primary care and health system contact, thus increasing the chances of lung cancer symptoms being identified and investigated earlier.

Small cell carcinoma is known for its aggressive nature and early metastasis [14], and thus it is not surprising that we found that small cell carcinoma, which was less prevalent

among immigrants, was associated with a later-stage diagnosis than adenocarcinoma. Nearly all cases of small cell carcinoma are associated with smoking [22]. Adenocarcinoma, which we found to be more prevalent among immigrants, is the most common subtype to be diagnosed in people who have never smoked (although smoking is still its main risk factor) [14,23]. Canadian immigrants are known to be generally less likely to smoke and more likely to quit than non-immigrants [24]. Importantly, smoking cessation support is a key component of Ontario's current lung cancer screening pilot [15].

This study has several limitations. First, cancer type was unknown for more than 28% of the study population, and thus, our findings regarding cancer type must be interpreted with caution. Second, immigrants who arrived before 1985 or who first migrated to another province before Ontario are not captured in the IRCC database and are thus included in the long-term resident group. Third, although we looked at the region of origin for immigrants, this obscures unique aspects about country of origin. For example, Newbold et al. found adjusted odds ratios of smoking as low as 0.30 (95% CI 0.26–0.36) for Jamaican-born immigrants and as high as 1.09 (95% CI 0.86–1.40) for Chilean-born immigrants versus Canadian-born [24]. In the current study, both of these countries would fall into the same region of Latin America and the Caribbean. Fourth, we were not able to capture any information about cigarette use, to which 90% of lung cancer cases can be attributed, or other risk factors for lung cancer such as exposure to asbestos, radon or second-hand smoke, or family history [14,25]. Finally, these results cannot be generalized to non-urban settings. Despite these limitations, our study is strengthened by its use of population-level high-quality demographic and healthcare data.

As lung cancer screening becomes more commonplace, future research should continue to monitor for inequalities in the stage of diagnosis and inequalities in screening rates. For Ontario's current organized screening programs (cervical, breast, and colorectal), immigrants and people living with low income have lower uptake [20,26–28], and lung screening programs may be susceptible to similar gaps. It is also crucial that there continue to be clear and efficient diagnostic pathways for people who do not quality for lung cancer screening but present with worrisome symptoms associated with lung cancer.

## 5. Conclusions

Lung cancer in Ontario is frequently diagnosed at a late stage. Immigrant status in Ontario is not associated with a late-stage diagnosis. However, men, people with fewer comorbidities, and people with small cell carcinoma have a higher likelihood of late-stage diagnosis. As lung cancer screening programs start to be implemented in Canada, targeted outreach to men and to smokers, increasing awareness about screening, and connecting every Canadian with primary care need to be system priorities.

**Author Contributions:** A.K.L. and G.D.D. conceived and designed the study and directed data analysis. E.G. and A.K. provided oversight to the analysis plan. H.L. analyzed the data. A.K.L., E.G., H.L., A.K., N.N.B., S.J.T.G., M.V. and G.D.D. interpreted the data. A.K.L. drafted the manuscript. A.K.L., E.G., H.L., A.K., N.N.B., S.J.T.G., M.V. and G.D.D. critically revised the manuscript for important intellectual content. H.L. had full access to all the data in the study and takes responsibility for the integrity of the data and the accuracy of the data analysis. All authors have read and agreed to the published version of the manuscript.

**Funding:** This study was supported by ICES, which is funded by an annual grant from the Ontario Ministry of Health (MOH) and the Ministry of Long-Term Care (MLTC). This study also received funding from the CIHR. Parts of this material are based on data and information compiled and provided by the MOH, Immigration, Refugees and Citizenship Canada (IRCC), Cancer Care Ontario (CCO), and the Canadian Institute for Health Information (CIHI). The analyses, conclusions, opinions, and statements expressed herein are solely those of the authors and do not reflect those of the funding or data sources; no endorsement is intended or should be inferred. A.L. is the Provincial Primary Care Lead, Cancer Screening at Ontario Health (Cancer Care Ontario). A.L. is supported by a CIHR New Investigator Award, as a Clinician Scientist by the University of Toronto Department of Family

and Community Medicine, and as Chair in Implementation Science at the Peter Gilgan Centre for Women's Cancers at Women's College Hospital, in partnership with the Canadian Cancer Society.

**Institutional Review Board Statement:** The study was conducted according to the guidelines of the Declaration of Helsinki and approved by the Unity Health Toronto Research Ethics Board (protocol code 19-072, approved on 2 April 2019).

**Informed Consent Statement:** Patient consent was waived. ICES is a prescribed entity under section 45 of Ontario's Personal Health Information Protection Act. Section 45 authorizes ICES to collect personal health information, without consent, for the purpose of analysis or compiling statistical information with respect to the management of, evaluation or monitoring of, the allocation of resources to or planning for all or part of the health system. This project was conducted under section 45, and approved by ICES' Privacy and Legal Office.

**Data Availability Statement:** The dataset from this study is held securely in coded form at ICES. While legal data sharing agreements between ICES and data providers (e.g., healthcare organizations and government) prohibit ICES from making the dataset publicly available, access may be granted to those who meet pre-specified criteria for confidential access, available at www.ices.on.ca/DAS, accessed on 6 Auguest 2019 (email: das@ices.on.ca). The full dataset creation plan and underlying analytic code are available from the authors upon request, understanding that the computer programs may rely upon coding templates or macros that are unique to ICES and are therefore either inaccessible or may require modification.

**Conflicts of Interest:** The authors have no conflict of interest to declare.

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
