# Peer review of "Lung Cancer Inequalities in Stage of Diagnosis in Ontario, Canada"

_curroncol, doi:10.3390/curroncol28030181_

Round 1

Reviewer 1 Report

Why do you think there is no difference between the immigrant and long term resident group? Perhaps provide some hypothesis or insight on the results.

The definition of long term residents is any immigrants who arrived before 1985.  That means immigrants can be living in Ontario from 0 to 32 years.  I think there must be differences between "recent" (say 0-5 years) and "longer term" immigrants (say >20 years).  For newer immigrants, they may be less likely to be familiar with the health care system, know the community resources available to them, and have support connections/networks than someone who pretty much lived in Ontario for a good portion of their life.  Is there anyway you can subcategorize the immigrant cohort to years of being in Ontario?

Language is one thing I am curious about.  What proportion of the immigrant cohort speak English proficiently as that will also affect their access to health care.

Your results highlight the criteria for lung cancer screening programs would be inequitable for immigrant patients as there is a higher proportion of younger patients and non-smokers in this group than the long term resident group. I think you should include a statement about that in your discussion.

Reviewer 2 Report

Very interesting study. Just a couple of things.

The tables particularly tables 1&2 are very large, can they be broken up or the data presented slightly differently? It is a bit difficult to synthesise the data in the tables as it is currently presented.

The uncertainty surrounding exact diagnosis due to the data not being available from the source databases is unfortunate.

Reviewer 3 Report

Thank you for giving me the opportunity to review this manuscript.

This study compares the stage at diagnosis among different categories of population. Although the results are not extrapolable to other populations outside of Ontario, this study is very interesting. It allows for better definition of populations to be targeted for prevention. In addition, the analysis is more thorough than most studies on the subject because it did not simply analyze all lung cancers in one group, but took into account the histological type.

I have no methodological criticism, the statistical analysis is very well done. The manuscript is very well written, the tables are easy to read. The discussion is interesting

Even if there is no major difference between the different groups. This study is interesting and deserves to be published.

I have minor comments :

I detect some inappropriate self citations  : To determine region of origin, we used country of birth and categorized each country into 143 a world region based on a published modification of the World Bank classification (11- 15). -> These references link to publications of the first author.
